# Quantification of Abdominal Muscle Mass and Diagnosis of Sarcopenia with Cross-Sectional Imaging in Patients with Polycystic Kidney Disease: Correlation with Total Kidney Volume

**DOI:** 10.3390/diagnostics12030755

**Published:** 2022-03-20

**Authors:** Chih-Horng Wu, Tai-Shuan Lai, Yung-Ming Chen, Chyi-Mong Chen, Shun-Chung Yang, Po-Chin Liang

**Affiliations:** 1Department of Medical Imaging, National Taiwan University Hospital, Taipei City 100, Taiwan; ared1020l@gmail.com (C.-H.W.); kimochen51@gmail.com (C.-M.C.); kind3000@ntuh.gov.tw (S.-C.Y.); 2Department of Internal Medicine, National Taiwan University Hospital, Taipei City 100, Taiwan; taishuanster@gmail.com (T.-S.L.); chenym@ntuh.gov.tw (Y.-M.C.); 3Department of Medical Imaging, National Taiwan University Hospital, Hsin-Chu Branch, Zhubei City 302, Taiwan

**Keywords:** sarcopenia, computed tomography, magnetic resonance imaging, polycystic kidney disease

## Abstract

Background: Polycystic kidney disease (PKD) is an inherited disorder characterized by renal cysts that may mask lean body loss. This study quantified and compared muscle mass by using computed tomography (CT) and magnetic resonance imaging (MRI) images between the PKD and control groups and correlated muscle mass with total kidney volume (TKV). Methods: We retrospectively enrolled patients who had a new diagnosis of PKD from May 2015 to May 2016. The CT and MRI images at the third lumbar level were processed to measure the total abdominal muscle (TAM) area for the diagnosis of sarcopenia, and TKV was estimated using the ellipsoid formula. Results: We included 37 women and 25 men (mean age: 50.40 years) in the PKD group. There was no difference in body mass index and albumin levels, but significant differences in creatinine level (*p* < 0.001), TAM area (*p* = 0.047), and TKV (*p* < 0.001), were noted between the two groups. A significantly negative correlation was observed between TKV and TAM area after adjustment for body height (*r* = −0.217, *p* = 0.003). Conclusions: CT and MRI images can accurately diagnose sarcopenia, which may be masked by cysts in patients with PKD.

## 1. Introduction

Muscle wasting occurs in patients with chronic kidney disease (CKD) and those undergoing renal replacement therapy because of accelerated protein catabolism [1,2]. A study reported that 20% of patients with end-stage kidney disease (ESKD) undergoing long-term hemodialysis experienced muscle mass loss [3]. Another study indicated that patients with muscle wasting had a poorer quality of life and prognosis than those without [4]. Body mass index (BMI), mid-upper arm circumference, and serum albumin level were previously adopted to evaluate nutrition status [5]. Recently, bioelectrical impedance analysis (BIA) and dual-energy X-ray absorptiometry (DXA) have been widely used to analyze body composition [6]. However, tissue edema, ascites, or dialysate may affect the findings of BIA and DXA. These factors limit the usage of BIA and DXA in patients with advanced cirrhosis or ESKD undergoing peritoneal dialysis [7,8,9].

Because of the presence of large and numerous cysts in the liver and kidneys of patients with polycystic kidney disease (PKD), BMI might mask their underweight status. Moreover, body composition can be divided into fat mass and fat-free mass, also known as lean body mass (LBM) [10]. LBM includes muscle and visceral proteins and predominantly consists of water, protein, glycogen, and minerals [11]. Therefore, the use of BIA and DXA findings might lead to the overestimation of LBM in patients with PKD [12]. By contrast, cross-sectional imaging modalities, such as computed tomography (CT) and magnetic resonance imaging (MRI), can quantitatively analyze the total abdominal muscle (TAM), visceral adipose tissue (VAT), and subcutaneous adipose tissue (SAT) after the exclusion of large and numerous cysts in the liver and kidneys through image processing [13].

Ubara et al. reported that renal artery embolization reduced the size of numerous cysts in the kidneys and the total kidney volume (TKV) [14]. Suwabe et al. demonstrated that renal artery embolization effectively improved quality of life and alleviated the symptoms of abdominal fullness and poor appetite [15]. However, Ubara et al. and a recent study reported that dry weight continuously decreased in the first 6 months and gradually increased thereafter [16]. These results are not in agreement with the finding of TKV reduction; this inconsistency can be explained by changes in body composition masked by the measurement of dry weight only.

Therefore, we hypothesize that a cross-sectional, image-based evaluation of muscle mass is a better indicator of malnutrition than BMI and serum biomarkers for patients with PKD. However, no study has quantified the muscle mass and adipose tissue in patients with PKD through CT and MRI. Therefore, this study aimed to measure TAM, VAT, and SAT areas at the third lumbar (L3) level with adjustment for body height (BH) and demonstrate the loss of muscle mass and adipose tissue in patients with PKD compared to the control group. In addition, this study determined the correlation of these measurements with TKV estimated using the ellipsoid formula.

## 2. Materials and Methods

### 2.1. Patient and Study Design

This retrospective, single-center, cross-sectional study enrolled adults who had a new diagnosis of PKD from May 2015 to May 2016 at National Taiwan University Hospital (NTUH). After the establishment of the diagnosis of PKD through a screening ultrasound based on Ravine’s criteria [17], abdominal CT or MRI was performed to evaluate PKD severity and TKV. Patients with insufficient clinical or laboratory data or inadequate imaging data for analysis were excluded. Potential liver donors and patients with a diagnosis of appendicitis during the same period were enrolled as controls. The clinical history of the control group was carefully reviewed to rule out the possibility of autoimmune disease, congestive heart failure, chronic lung disease, cirrhosis, and CKD. For each patient with PKD, two age- and sex-matched controls were included. The associated clinical, laboratory, and imaging data were collected within 1 month for both the PKD and control groups. Figure 1 presents the flowchart of patient enrollment. This study adhered to the tenets of the Declaration of Helsinki and was approved by the Institutional Review Board of NTUH (approval no.: 202006139RINA). The requirement of informed consent was waived because of the retrospective nature of this study, which posed no additional risk to patients.

### 2.2. Study Variables and Imaging Acquisition

Clinical data (age, sex, body weight (BW), BH, and BMI) and laboratory data (albumin and creatinine levels and estimated glomerular filtration rate (eGFR)) were collected from electronic medical records. The BMI classifications were underweight (under 18.5 kg/m^2^), normal weight (18.5 to 24.9 kg/m^2^), overweight (25 to 29.9 kg/m^2^), and obese (30 kg/m^2^ and above). The stages of CKD were based on the eGFR and were classified as follows: stage 1: >90, stage 2: 60–89, stage 3: 30–59, stage 4: 15–29, and stage 5: <15 mL/min/1.73 m^2^.

If patients had no contraindication for MRI (e.g., claustrophobia, metal implants, or pacemakers) and were able to suspend respiration for image acquisition, we performed MRI because of the absence of radiation exposure during the examination. For MRI, we used a 3.0-Tesla unit (Magnetom Verio; Siemens Medical Solutions, Erlangen, Germany). We performed coronal T2 half-Fourier acquisition single-shot turbo spin-echo imaging with a repetition time (TR) of 2000 ms, time to echo (TE) of 92 ms, and a slice thickness of 3 mm for the abdomen and pelvis to evaluate the cyst extension and determine TKV [18]. We used a triple-echo gradient-echo sequence with a water and fat separation technique based on the Dixon method for the abdomen and pelvis to evaluate fat mass and fat-free mass with the following parameters: TR: 9.65 ms/TE: 2.45 (IP1), 3.67 (OP), and 7.35 (IP2) ms; flip angle: 12°; and slice thickness: 5 mm. The T2* difference, water–fat ambiguity, and B0 inhomogeneity were corrected by following the technique reported in a previous study [19], and a fat-signal fraction map was generated.

A CT scan was performed only in patients with contraindications for MRI or who were unable to suspend respiration. We used a 64-slice CT scanner (Lightspeed VCT; GE Medical System, Waukesha, WI, USA) with the following parameters: 120 kV dose modulation according to body size; and 5 mm slice thickness. We obtained only nonenhanced images to prevent contrast-induced nephropathy.

### 2.3. Imaging Processing

TKV was estimated through multiplanar reformation and measured using the IntelliSpace Portal 9.0 workstation (Philips, Amsterdam, The Netherlands). The maximal longitudinal length (L) of the kidney was determined on T2-weighted tilted coronal slices parallel to the long renal axis (Figure 2D). The maximal width (W) was determined perpendicular to the L in the same plane in which the L was localized (Figure 2C). Finally, the maximal depth (D) was determined perpendicular to the L in a thick sagittal slice (Figure 2D) [18]. TKV was estimated as follows:(1)$$\mathrm{Estimated}\;\mathrm{TKV}\;\left(\mathrm{mL}\right)=\frac{\pi }{6}\times L\times W\times D$$

Nonenhanced MRI and CT images at the L3 level were analyzed to determine TAM, VAT, and SAT areas processed on a compatible computer by using open-source software (ImageJ version 1.51; National Institutes of Health, Bethesda, MD, USA). Contours were obtained using a manual tracing method (Figure 3). A radiologist (C.-H.W.) with 10 years of experience in abdominal imaging processed the images based on the method described previously [20]. The TAM, VAT, and SAT indices were calculated as follows: TAM, VAT, and SAT area/(BH [m])^2^. Sarcopenia was defined as a TAM index of <52.4 cm^2^/m^2^ in men and <38.5 cm^2^/m^2^ in women, according to Prado’s protocol [21].

### 2.4. Statistical Analysis

Data were analyzed using the following tools: Excel 2007 (Microsoft, Redmond, WA, USA) and R 3.4.3 (R Foundation for Statistical Computing, Vienna, Austria). A *p*-value of <0.05 was considered statistically significant. Categorical and continuous variables were compared between the PKD and control groups by using the chi-square test and Student’s t test, respectively. In addition, Pearson’s correlation coefficient was calculated to determine the correlation of TAM, VAT, and SAT areas and indices with estimated TKV. Logistic regression was conducted in the univariate and multivariate analysis to predict the presence of sarcopenia. We determined the required sample size to be approximately 48 in the PKD group and 96 in the control group for continuous variables, assuming a significance of 5% (α = 0.05), a power of 80% (β = 0.20), a mean of 40 cm^2^/m^2^ with a standard deviation of 10 cm^2^/m^2^ for the PKD group, and a mean of 45 cm^2^/m^2^ with a standard deviation of 10 cm^2^/m^2^ for the control group.

## 3. Results

### 3.1. Clinical and Laboratory Assessment

We included 37 women and 25 men with a mean age of 50.40 (24–70) years in the PKD group and 74 women and 50 men with a mean age of 50.18 (24–70) years in the control group. No differences in BW, BH, BMI, serum albumin, liver function tests and lipid profiles were noted between the PKD and control groups. However, significant differences in the serum creatinine level and eGFR were observed between the PKD and control groups. Among the 62 patients with PKD, 15 had stage 1 CKD, 25 had stage 2 CKD, 9 had stage 3 CKD, and 13 had stage 4 or 5 CKD. Furthermore, 10 patients had no hepatic cysts, and 52 patients had hepatic cysts. The estimated TKV was significantly larger in the PKD group than in the control group (mean: 1869 vs. 309 mL, *p* < 0.001). In addition, a higher proportion of the patients in the PKD group had sarcopenia (69.4%, 43/62) than did those in the control group (40.3%, 50/124; Table 1).

### 3.2. Correlation between Body Composition Analysis and TKV

The PKD group had a significantly lower TAM area, VAT area, TAM index, and VAT index (mean: 111.12 vs. 122.23 cm^2^, 77.20 vs. 108.58 cm^2^, 39.65 vs. 45.68 cm^2^/m^2^, and 27.62 vs. 40.15 cm^2^/m^2^; *p* = 0.047, 0.013, <0.001, and 0.004, respectively) than the control group. The SAT area and index did not significantly differ between the groups (125.94 vs. 134.81 cm^2^ and 46.23 vs. 51.20 cm^2^/m^2^; *p* = 0.396 and 0.189, respectively; Table 1). A negative correlation was observed between TKV and the TAM index (*r* = −0.217, *p* = 0.003); however, no correlation was noted between TKV and the VAT or SAT indices (Table 2).

### 3.3. Sarcopenia Assessment

The findings of the univariate analysis indicated that age, BMI, creatinine levels, and estimated TKV were significant factors associated with the presence of sarcopenia (TAM index of <52.4 cm^2^/m^2^ in men and <38.5 cm^2^/m^2^ in women). In the multivariate analysis, BMI and estimated TKV were still significant factors associated with the presence of sarcopenia (odds ratio: 0.795 and 1.001; 95% confidence interval: 0.721–0.877 and 1.000–1.001, respectively) (Table 3).

### 3.4. Internal Validation

To validate the reproducibility of body composition measurements, TAM, VAT, and SAT areas were measured by another radiologist (P.C.L) with 20 years of experience in abdominal radiology. The concordance correlation coefficient between reader 1 (C.H.W) and reader 2 (P.C.L) was 0.988 for the TAM area, 0.996 for the VAT area, and 0.997 for the SAT area (all *p* < 0.001). For the sarcopenia assessment, the kappa value was 0.888 (*p* < 0.001), indicating very low interobserver variability and satisfactory agreement between reader 1 and reader 2.

## 4. Discussion

PKD is the fourth-leading cause of ESKD in the United States, where 33 patients per million have to initiate dialysis every year due to disease progression [22]. In the late stages of PKD, cysts replace the renal parenchyma and cause renal function deterioration and renomegaly [23]. Our results demonstrated that cysts with a large volume and high weight masked BW and BMI changes in the PKD group. There were no differences in BMI categories, serum albumin and lipid profiles between the PKD and control groups. Therefore, muscle loss cannot be evaluated by clinical and laboratory data in our study. Similarly, DXA cannot separate fat and fat-free mass containing renal and hepatic cysts. Therefore, only cross-sectional images can differentiate among the TAM, VAT, SAT, and cysts and demonstrate TAM and VAT loss in patients with PKD, as our results showed.

Because muscle and fat mass are related to age and sex, age- and sex-matched controls should be included for comparison with patients with PKD. Liver donors and patients with appendicitis, who closely represent the normal population, with CT or MRI images were included as controls in this study. Although our control group might not have had the same characteristics as the normal population, our goal was to evaluate whether CT and MRI can serve as better tools than BW and BMI for patients with PKD. Our results revealed no difference in the SAT area but significant differences in TAM and VAT areas between the PKD and control groups. This finding indicates that the measurement of the SAT or other subcutaneous physical parameters might not be sufficient to detect poor nutrition in patients with PKD. PKD might mainly affect the TAM and VAT.

Our study demonstrated that the TAM index was negatively correlated with the estimated TKV but not with VAT and SAT. A previous study reported that protein loss and poor appetite might cause sarcopenia in patients with CKD [24]. Therefore, the mass effect of renomegaly may be more associated with protein catabolism. In clinical practice, BMI and serum creatinine may play a crucial role in muscle loss for patients with CKD [25]. However, most CKD patients do not have renomegaly to the same degree as patients with PKD. Our study demonstrated that TKV was also a significant determining factor for sarcopenia. Therefore, sarcopenia in patients with PKD was not only related to low BMI but also large renal cysts. In PKD patients, the increase in creatinine was challenging to reverse but the TKV might be controlled by cyst reduction therapy. Therefore, we pointed out the role of TKV for further study and research.

This study has several limitations. Firstly, this single-center study included a moderate sample size. However, we included more cases above our statistical hypothesis with 0.05 in type I and 0.20 in type II errors. Additional large-scale and follow-up longitudinal studies should be conducted using our method to evaluate body composition and sarcopenia in patients with PKD. Secondly, there was only a weak correlation (*r* = −0.217) between the TKV and muscle (TAM) index, but no correlation was observed between the TKV and adipose tissue (VAT and SAT). Therefore, the mechanism underlying sarcopenia should be explored. Muscle biopsy is not routinely performed in patients with sarcopenia. Therefore, noninvasive or less invasive studies, such as perfusion MRI, magnetic resonance spectroscopy, serum-amino-acid-level examination, and genetic analysis, are required to explore the etiology of sarcopenia in PKD. Finally, our measurements still relied on an experienced radiologist to evaluate large cysts in the liver and kidneys. However, we performed internal validation by including two experienced abdominal radiologists. The results indicated that our method used to evaluate TAM, VAT, and SAT areas in patients with PKD was highly reproducible with a small interobserver variability; however, automatic imaging tools should be developed to detect sarcopenia in these patients [26].

## 5. Conclusions

In conclusion, the abdominal muscle mass loss in patients with PKD could be accurately evaluated and diagnosed through cross-sectional imaging without being masked by BMI. The kidney volume is negatively correlated with abdominal muscle mass, not adipose tissue. In addition to age, BMI, and serum creatinine, kidney volume is also an important indicator for sarcopenia in patients with PKD.

## Figures and Tables

**Figure 1 diagnostics-12-00755-f001:**
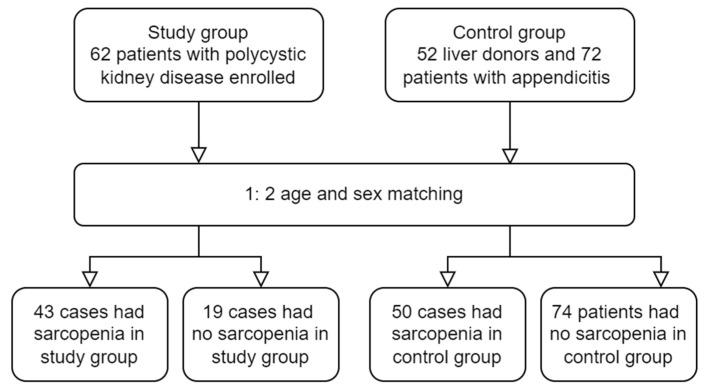
Flowchart of patient enrolment.

**Figure 2 diagnostics-12-00755-f002:**
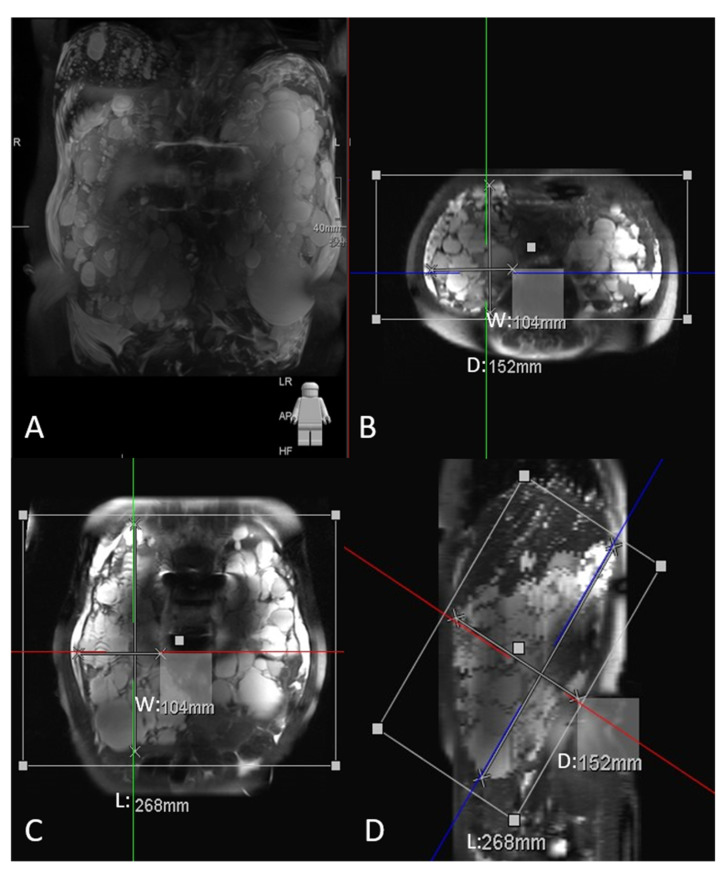
Multiplanar reformation can estimate total kidney volume with the maximal intensity projection (**A**). The maximal length (L), width (W), and depth (D) can be obtained from the oblique coronal plane (**C**), sagittal plane (**D**), and tilted axial plane (**B**), sequentially.

**Figure 3 diagnostics-12-00755-f003:**
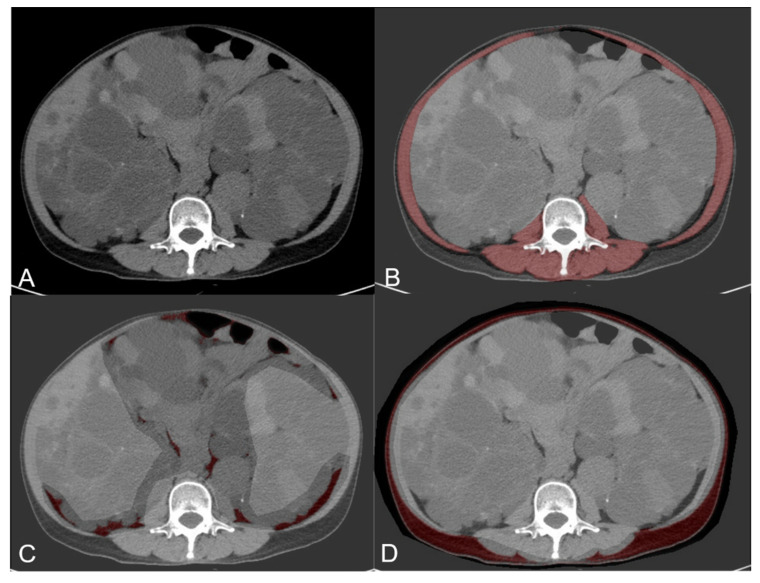
The nonenhanced CT image (**A**) at the L3 level was analyzed to determine the TAM (**B**), VAT (**C**), and SAT (**D**) area by using ImageJ. CT: computed tomography, TAM: total abdominal muscle, VAT: visceral adipose tissue, SAT: subcutaneous adipose tissue.

**Table 1 diagnostics-12-00755-t001:** Demographic data of patients with polycystic kidney disease and controls (52 potential liver donors and 72 patients with appendicitis).

Group (n)	PKD (62)	Control (124)	*p* Value
Age (years)	50.40 ± 12.20	50.18 ± 12.09	0.905
Female/Male	37/25	74/50	0.999
BW (kg)	64.80 ± 17.08	63.84 ± 15.32	0.697
BH (cm)	164.17 ± 9.68	162.53 ± 8.02	0.223
BMI (kg/m^2^)	23.75 ± 4.27	23.97 ± 4.31	0.742
Underweight/Normal/ Overweight/Obese	6/35/16/5	10/162/51/25	0.472
Albumin (g/dL)	4.4 ± 0.3	4.4 ± 0.6	0.686
Bilirubin (mg/dL)	0.71 ± 0.25	0.75 ± 0.40	0.522
AST (U/L)	21 ± 8	20 ± 9	0.832
Platelet (K/µL)	212 ± 47	223 ± 68	0.323
Triglyceride (mg/dL)	111 ± 50	109 ± 66	0.917
Total cholesterol (mg/dL)	173 ± 30	178 ± 35	0.513
Creatinine (mg/dL)	2.32 ± 2.95	0.78 ± 0.22	<0.001 *
eGFR (mL/min/1.73 m^2^)	66.04 ± 36.72	101.00 ± 23.53	<0.001 *
Estimated TKV (mL)	1869 ± 1967	309 ± 76	<0.001 *
TAM area (cm^2^)	111.12 ± 37.85	122.23 ± 34.65	0.047 *
VAT area (cm^2^)	77.20 ± 67.62	108.58 ± 101.24	0.013 *
SAT area (cm^2^)	125.94 ± 60.84	134.81 ± 69.86	0.396
TAM index (cm^2^/m^2^)	39.65 ± 11.44	45.68 ± 10.05	<0.001 *
VAT index (cm^2^/m^2^)	27.62 ± 22.20	40.15 ± 35.57	0.004 *
SAT index (cm^2^/m^2^)	46.23 ± 20.77	51.20 ± 25.73	0.189
TAM-defined sarcopenia	43	50	<0.001 *

Represented with number or mean ± standard deviation. Abbreviations: PKD: polycystic kidney disease, BW: body weight, BH: body weight, BMI: body mass index, eGFR: estimated glomerular filtration rate, *: *p* value < 0.05.

**Table 2 diagnostics-12-00755-t002:** Correlation between body composition analysis and total kidney volume.

Correlation with Estimated TKV	Correlation Coefficient (*r*)	*p* Value
TAM area (cm^2^)	−0.031	0.679
VAT area (cm^2^)	−0.007	0.924
SAT area (cm^2^)	−0.049	0.503
TAM index (cm^2^/m^2^)	−0.217	0.003 *
VAT index (cm^2^/m^2^)	−0.027	0.714
SAT index (cm^2^/m^2^)	−0.097	0.186

Abbreviations: TKV: total kidney volume, TAM: total abdominal muscle, VAT: visceral adipose tissue, SAT: subcutaneous adipose tissue, *: *p* value < 0.05.

**Table 3 diagnostics-12-00755-t003:** Univariate and multivariate analysis of factors associated with sarcopenia (TAM index of <52.4 cm^2^/m^2^ in men and <38.5 cm^2^/m^2^ in women).

	Univariate Analysis			Multivariate Analysis		
Factors	OR	95% CI	*p*	OR	95% CI	*p*
Age (year)	1.025	1.000–1.050	0.047 *	1.027	0.999–1.055	0.055
Gender	1.796	0.993–3.249	0.395			
BMI (kg/m^2^)	0.854	0.787–0.926	<0.001 *	0.795	0.721–0.877	<0.001 *
Albumin (g/dL)	0.935	0.361–2.422	0.890			
Bilirubin (mg/dL)	0.900	0.325–2.494	0.839			
Creatinine (mg/dL)	1.389	1.041–1.854	0.025 *	1.054	0.728–1.527	0.779
eGFR (mL/min/1.73 m^2^)	1.043	0.994–1.096	0.089			
Estimated TKV (mL)	1.001	1.000–1.001	0.004 *	1.001	1.000–1.001	0.013 *

Abbreviations: TAM: total abdominal muscle, OR: odds ratio, CI: confidence interval, BMI: body mass index, eGFR: estimated glomerular filtration rate, TKV: total kidney volume, *: *p*-value < 0.05.

## Data Availability

Not applicable.

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
