# Peer review of "Quantification of Abdominal Muscle Mass and Diagnosis of Sarcopenia with Cross-Sectional Imaging in Patients with Polycystic Kidney Disease: Correlation with Total Kidney Volume"

_diagnostics, 2022, doi:10.3390/diagnostics12030755_

Round 1

Reviewer 1 Report

  1. The aim of the study should be expressed more precisely.
  2. “A negative correlation was observed between TKV and the TAM index (r = −0.217, p = 0.003)”. Presented correlation, however significant, is rather weak (-0.217) and do not express the real correlation between two variables.
  3. Authors stated that „BMI and estimated TKV were still significant factors associated with the presence of sarcopenia” but on the other hand the PKD patients presented worst kidney function, as expressed by serum creatinine concentration, what is a crucial factor of malnutrition and therefore sarcopenia. How could Authors explain that?

Author Response

We appreciate very much the reviewers’ comments regarding our manuscript entitled “Quantification of Abdominal Muscle Mass and Diagnosis of Sarcopenia With Cross-Sectional Imaging in Patients With Polycystic Kidney Disease: Correlation with Total Kidney Volume” (diagnostics-1637418). The following is a point-by-point response to the reviewers’ comments, with according manuscript revision.

Reviewer #1:

Point 1: The aim of the study should be expressed more precisely.

Response 1:

Many thanks to the reviewer. We re-emphasized our goal in this study and explained it in more detail in the last paragraph of the introduction.

We revised our manuscript as below.

(page 2, 1. Introduction)

Therefore, this study aimed to measure TAM, VAT, and SAT areas at the third lumbar (L3) level with adjustment of body height (BH) and to demonstrate the loss of muscle mass and adipose tissue in patients with PKD compared to the control group. In addition, this study also determined the correlation of these measurements with TKV estimated using the ellipsoid formula.s.

Point 2: “A negative correlation was observed between TKV and the TAM index (r = −0.217, p = 0.003)”. Presented correlation, however significant, is rather weak (-0.217) and do not express the real correlation between two variables.

Response 2:

Many thanks to the reviewer. We agreed that although the p value was significant, there was only a weak correlation between TKV and TAM. However, we wanted to present that the TKV was more associated with the muscle mass (TAM) than the adipose tissue (VAT and SAT). We also put this point in the limitation in the last paragraph of the discussion.

We revised our manuscript as below.

(page 8, 4. Discussion)

Second, there was only a weak correlation (r = −0.217) between the TKV and muscle (TAM) index but no correlation was observed between the TKV and adipose tissue (VAT and SAT). Therefore, the mechanism underlying sarcopenia should be explored. Muscle biopsy is not routinely performed in patients with sarcopenia. Therefore, noninvasive or less invasive studies such as perfusion MRI, magnetic resonance spectroscopy, serum amino acid level examination, and genetic analysis are required to explore the etiology of sarcopenia in PKD.

Point 3: Authors stated that BMI and estimated TKV were still significant factors associated with the presence of sarcopenia” but on the other hand the PKD patients presented worst kidney function, as expressed by serum creatinine concentration, what is a crucial factor of malnutrition and therefore sarcopenia. How could Authors explain that?

Response 3:

Many thanks to the reviewer. We agreed that serum creatinine is a crucial factor for malnutrition in patients with chronic kidney disease (CKD). However, most of the CKD patients don’t have renomegaly as patients with PKD. Therefore, BMI and serum creatinine are important factors in developing sarcopenia in CKD patients, but the TKV is an added value in determining sarcopenia in PKD patients. Furthermore, in PKD patients, the increase of creatinine is difficult to reverse but the TKV might be controlled by cyst reduction therapy. Therefore, we pointed out the role of TKV for further study and research. We further emphasized these points in the 3rd paragraph of the discussion.

We revised our manuscript as below.

(page 7, 4. Discussion)

Our study demonstrated that the TAM index was negatively correlated with estimated TKV but not with VAT and SAT. A previous study reported that protein loss and poor appetite might cause sarcopenia in patients with CKD [24]. Therefore, the mass effect of renomegaly may be more associated with protein catabolism. In clinical practice, BMI and serum creatinine may play a crucial role in muscle loss for patients with CKD [25]. However, most CKD patients don’t have renomegaly as patients with PKD. Our study demonstrated that TKV was also a significant determining factor for sarcopenia. Therefore, sarcopenia in patients with PKD was not only related to low BMI but also large renal cysts. In PKD patients, the increase of creatinine is challenging to reverse but the TKV might be controlled by cyst reduction therapy. Therefore, we pointed out the role of TKV for further study and research.

Reviewer 2 Report

The item is well built.
In the introduction part, the justification of the objective must be improved, that is, how the study hypothesis is stated.
As for the sample, it is small, but it can be enough.
In the methodological part, it must justify the characteristics of the control group.
In the discussion part, you must improve a justification that explains the results, that is, why you obtain those results.
The conclusions can be a little broader.

Author Response

Reviewer #2:

Point 1: In the introduction part, the justification of the objective must be improved, that is, how the study hypothesis is stated.

Response 1:

Thanks for the suggestion from the reviewer. We further described the hypothesis and goal of our study in the last paragraph of the introduction.

We revised our manuscript as below.

(page 2, 1. Introduction)

Therefore, we hypothesize a cross-sectional image-based evaluation of muscle mass is a better indicator for malnutrition than BMI and serum biomarkers for patients with PKD. However, no study has quantified the muscle mass and adipose tissue in patients with PKD through CT and MRI. Therefore, this study aimed to measure TAM, VAT, and SAT areas at the third lumbar (L3) level with adjustment of body height (BH) and to demonstrate the loss of muscle mass and adipose tissue in patients with PKD compared to the control group. In addition, this study also determined the correlation of these measurements with TKV estimated using the ellipsoid formula.

Point 2: As for the sample, it is small, but it can be enough

Response 2:

Thanks for the comment from the reviewer. We have described the sample size estimation in 2.4 Statistical Analysis of the Materials and Methods and presented in the limitation of the Discussion.

Point 3: In the methodological part, it must justify the characteristics of the control group

Response 3:

Thanks for the suggestion from the reviewer. We added more characteristics in the demographic data (Table 1) of the control group. There was no difference in BMI, serum albumin, liver function tests, and lipid profiles between the PKD and control groups.

We revised our manuscript as below:

(page 5, Clinical and laboratory assessment in Results)

No differences in BW, BH, BMI, serum albumin, liver function tests and lipid profiles were noted between the PKD and control groups. However, a significant difference in the serum creatinine level and eGFR was observed between the PKD and control groups.

(page 6, Table 1)

Table 1. Demographic data of patients with polycystic kidney disease and controls.

(52 potential liver donors and 72 patients with appendicitis).

Group (n)

PKD (62)

Control (124)

P value

Age (years)

50.40 ± 12.20

50.18 ± 12.09

0.905

Female/Male

37/25

74/50

0.999

BW (kg)

64.80 ± 17.08

63.84±15.32

0.697

BH (cm)

164.17 ± 9.68

162.53 ± 8.02

0.223

BMI (kg/m2)

23.75 ± 4.27

23.97 ± 4.31

0.742

Underweight/Normal/

Overweight/Obese

6/35/16/5

10/162/51/25

0.472

Albumin (g/dL)

4.4 ± 0.3

4.4 ± 0.6

0.686

Bilirubin (mg/dL)

0.71 ± 0.25

0.75 ± 0.40

0.522

AST (U/L)

21 ± 8

20 ± 9

0.832

Platelet (K/µL)

212 ± 47

223 ± 68

0.323

Triglyceride (mg/dL)

111 ± 50

109 ± 66

0.917

Total cholesterol (mg/dL)

173 ± 30

178 ± 35

0.513

Creatinine (mg/dL)

2.32 ± 2.95

0.78 ± 0.22

<0.001*

eGFR (mL/min/1.73m2)

66.04 ± 36.72

101.00 ± 23.53

<0.001*

Estimated TKV (ml)

1869 ± 1967

309 ± 76

<0.001*

TAM area (cm2)

111.12 ± 37.85

122.23 ± 34.65

0.047*

VAT area (cm2)

77.20 ± 67.62

108.58 ± 101.24

0.013*

SAT area (cm2)

125.94 ± 60.84

134.81 ± 69.86

0.396

TAM index (cm2/m2)

39.65 ± 11.44

45.68 ± 10.05

<0.001*

VAT index (cm2/m2)

27.62 ± 22.20

40.15 ± 35.57

0.004*

SAT index (cm2/m2)

46.23 ± 20.77

51.20 ± 25.73

0.189

TAM-defined sarcopenia

43

50

<0.001*

Represented with number or mean ± standard deviation. Abbreviations: PKD: polycystic kidney disease, BW: body weight, BH: body weight, BMI: body mass index, eGFR: estimated glomerular filtration rate, *: P value <0.05.

Point 4: In the discussion part, you must improve a justification that explains the results, that is, why you obtain those result.

Response 4:

Thanks for the suggestion from the reviewer. We added more explanation of our data in the Discussion. The 1st paragraph emphasized no difference in nutrition surveys in clinical and laboratory examinations except renal function. Only abdominal CT or MRI images demonstrate abdominal muscle and visceral adipose tissue loss in patients with PKD. Then we also explained the negative correlation between the abdominal muscle mass and TKV in the 3rd paragraph. The BMI and serum creatinine may be good indicators to predict sarcopenia in patietns with CKD. However, the renomegaly in patients with PKD causes TKV might be another indicator to predict sarcopenia. Finally, we reveled the limitation of our study, including sample size, further potential surveys about sarcopenia, including muscle perfusion, quality and even biopsy.

We revised our manuscript as below:

(page 7-8, Discussion)

PKD is the fourth leading cause of ESKD in the United States, where 33 patients per million have to initiate dialysis due to disease progression every year [22]. In the late stages of PKD, cysts replace the renal parenchyma and cause renal function deterioration and renomegaly [23]. Our results demonstrated that cysts with large volume and high weight masked BW and BMI changes in the PKD group. There was no difference in BMI categories, serum albumin and lipid profiles between the PKD and control groups. Therefore, muscle loss cannot be evaluated by clinical and laboratory data in our study. Similarly, DXA cannot separate fat and fat-free mass containing renal and hepatic cysts. Therefore, only cross-sectional images can differentiate among the TAM, VAT, SAT, and cysts and demonstrate TAM and VAT loss in patients with PKD, as our result showed.

Because the muscle and fat mass are related to age and sex, age- and sex-matched controls should be included for comparison with patients with PKD. Liver donors and patients with appendicitis, who closely represent the normal population, with CT or MRI images were included as controls in this study. Although our control group might not have the same characteristics as the normal population, our goal was to evaluate whether CT and MRI can serve as better tools than BW and BMI for patients with PKD. Our results revealed no difference in the SAT area but a significant difference in TAM and VAT areas between the PKD and control groups. This finding indicates that the measurement of the SAT or other subcutaneous physical parameters might not be sufficient to detect poor nutrition in patients with PKD. PKD might mainly affect the TAM and VAT.

Our study demonstrated that the TAM index was negatively correlated with estimated TKV but not with VAT and SAT. A previous study reported that protein loss and poor appetite might cause sarcopenia in patients with CKD [24]. Therefore, the mass effect of renomegaly may be more associated with protein catabolism. In clinical practice, BMI and serum creatinine may play a crucial role in muscle loss for patients with CKD [25]. However, most CKD patients don’t have renomegaly as patients with PKD. Our study demonstrated that TKV was also a significant determining factor for sarcopenia. Therefore, sarcopenia in patients with PKD was not only related to low BMI but also large renal cysts. In PKD patients, the increase of creatinine is challenging to reverse but the TKV might be controlled by cyst reduction therapy. Therefore, we pointed out the role of TKV for further study and research.

This study has several limitations. First, this single-center study included a moderate sample size. However, we included more cases above our statistical hypothesis with 0.05 in type I and 0.20 in type II errors. Additional large-scale and follow-up longitudinal studies should be conducted using our method to evaluate body composition and sarcopenia in patients with PKD. Second, there was only a weak correlation (r = −0.217) between the TKV and muscle (TAM) index but no correlation was observed between the TKV and adipose tissue (VAT and SAT). Therefore, the mechanism underlying sarcopenia should be explored. Muscle biopsy is not routinely performed in patients with sarcopenia. Therefore, noninvasive or less invasive studies such as perfusion MRI, magnetic resonance spectroscopy, serum amino acid level examination, and genetic analysis are required to explore the etiology of sarcopenia in PKD. Finally, our measurements still relied on an experienced radiologist to evaluate large cysts in the liver and kidneys. However, we performed internal validation by including two experienced abdominal radiologists. The result indicated that our method used to evaluate TAM, VAT, and SAT areas in patients with PKD was highly reproducible with small interobserver variability. Although our protocol was highly reproducible with small interobserver variability, automatic imaging tools should be developed to detect sarcopenia in these patients [26].

Point 5: The conclusions can be a little broader.

Response 5:

Thanks for the suggestion from the reviewer. We added two significant findings in our study in the conclusion

We revised our manuscript as below:

(page 8, Conclusion)

In conclusion, the abdominal muscle mass loss in patients with PKD could be accurately evaluated and diagnosed through cross-sectional imaging without being masked by BMI. The kidney volume is negatively correlated with abdominal muscle mass, not adipose tissue. In addition to age, BMI and serum creatinine, kidney volume is also an important indicator for sarcopenia in patients with PKD.

Round 2

Reviewer 1 Report

Thank you for the answers provided. I do not have any further remarks.